# In Vivo Emergence of Podovirus Resistance via *tarS* Mutation During Phage-Antibiotic Treatment of Experimental MSSA Endocarditis

**DOI:** 10.3390/v17081039

**Published:** 2025-07-25

**Authors:** Jérémy Cherbuin, Jonathan Save, Emma Osswald, Grégory Resch

**Affiliations:** 1Laboratory of Bacteriophages and Phage Therapy, Center for Research and Innovation in Clinical Pharmaceutical Sciences (CRISP), Lausanne University Hospital (CHUV), CH-1011 Lausanne, Switzerland; jeremy.cherbuin@unibe.ch (J.C.);; 2Institute of Veterinary Bacteriology, Department of Infectious Diseases and Pathobiology, Vetsuisse Faculty, University of Bern, CH-3012 Bern, Switzerland

**Keywords:** bacteriophage, phage therapy, phage cocktail, phage-antibiotic synergism, *Staphylococcus aureus*, experimental infective endocarditis, flucloxacillin

## Abstract

Phage therapy shows promise as an adjunct to antibiotics for treating Staphylococcus aureus infections. We previously reported a combined flucloxacillin/two-phage cocktail treatment selected for resistance to podovirus phage 66 in a rodent model of methicillin-susceptible *S. aureus* (MSSA) endocarditis. Here we show that resistant clones harbor mutations in *tarS*, which encodes a glycosyltransferase essential for β-GlcNAcylation of wall teichoic acid (WTA). This WTA modification has been described in vitro as critical for *podoviruses* adsorption. Transcriptomics confirmed continued *tarS* expression in resistant clones, supporting a loss-of-function mechanism. Accordingly, phage 66 binding and killing were restored by WT *tarS* complementation. In addition, we investigated the counterintuitive innate susceptibility to phage 66 of the *tarM* + Laus102 strain used in the endocarditis model. We show that it likely results from a significant lower *tarM* expression, in contrast to the innate resistant strain RN4220. Our findings demonstrate that *tarS*-mediated WTA β-GlcNAcylation is critical for podovirus infection also in vivo and identify *tarM* transcriptional defect as a new mechanism of podoviruses susceptibility in *S. aureus*. Moreover, and since *tarS* disruption has been previously shown to enhance β-lactam susceptibility, our results support the development of combined podovirus/antibiotic strategies for the management of MRSA infections.

## 1. Introduction

The overuse of antibiotics has created such intense selection pressure that previously mild bacterial infections, especially hospital-acquired ones, are becoming harder to treat with standard therapies [1]. In this alarming context, one innovative approach gaining renewed interest in the West is phage therapy—the therapeutic use of viruses that infect bacteria, known as bacteriophages [2]. Phages were discovered and used therapeutically nearly a decade before penicillin but began to vanish from Western medicine by the 1960s. Their decline stemmed largely from the complexity of personalized treatment, which contrasted with the broad, simple application of antibiotics [3]. For phage therapy to be widely adopted, a deep understanding of phage–bacteria interactions is essential. Under phage pressure, bacteria can acquire mutations or anti-phage defenses to evade death [4]. In turn, phages evolve to bypass these defenses, creating an ongoing “arms race” in nature [5]. The first step of the phage life cycle is adsorption to the bacterial cell wall. In *S. aureus*, phages often bind to wall teichoic acids (WTA) on the bacterial surface [6]. Accordingly, *S. aureus* can resist phage infection in vitro by altering WTA, specifically by removing a β-N-acetylglucosamine (β-GlcNAc) residue, which prevents certain phages from attaching [7]. In response, phages may develop point mutations in their receptor binding proteins (RBPs), enabling them to recognize modified WTA or alternative receptors. For example, the Twort-like phage ΦSA012 has two RBPs that bind either glycosylated WTA or the WTA backbone itself [8]. In *S. aureus*, WTA consists of ribitol-5-phosphate (RboP) units linked to the peptidoglycan via glycerol phosphate (GroP) [9]. Beyond its role in cell wall integrity, WTA is crucial to *S. aureus* pathogenesis—impacting nasal colonization, immune interaction, β-lactam resistance, and abscess formation [10,11]. The *tar* (and *tag*) genes govern WTA biosynthesis [10,12]. Specifically, *tarM*, *tarS*, and *tarP* encode glycosyltransferases that modify WTA before export. TarM performs α-glycosylation at C4, TarS β-glycosylates the same site, and TarP β-glycosylates at C3 [13]. These glycosylation patterns determine phage binding. For instance, strains with *tarM* and O-linked-α-N-acetylglucosaminylated (α-O-GlcNAcylated) WTA, such as RN4220 and USA300, resist infection by podoviruses. In contrast, exclusive β-glycosylation via *tarS* or *tarP* allows infection to occur [14]. Although clinically significant, in vivo development of phage-resistant *S. aureus* during phage therapy is still underexplored [15,16]. We therefore analyzed resistant *S. aureus* mutants previously isolated from cardiac vegetations in rats with MSSA infective endocarditis. These animals had been treated with a two-phage cocktail (myovirus phage vB_SauH_2002 and podovirus phage 66) in combination with flucloxacillin, a standard of care antibiotic [17].

## 2. Material and Methods

### 2.1. Bacterial and Phage Strains and Growth Conditions

*S. aureus* strain Laus102 is a methicillin-susceptible strain (MSSA) isolated from a healthy human [18]. RN4220 (hla-, derivative of *S. aureus* 8325-4) was obtained from O’Reilly and Novick [19]. Strains were cultured in tryptic soy broth (TSB) and tryptic soy agar (TSA) plates at 37 °C. *Herelleviridae* phage vB_SauH_2002 was isolated from sewage [17], while *Rountreeviridae* phage 66 was purchased from the National Collection of Types Culture of Public Health England (NCTC, https://www.culturecollections.org.uk/, accessed on 1 January 2022). *E. coli* DH10B was used for cloning and grown in Lysogeny Broth (LB) or on LB-agar (LA) at 37 °C. Antibiotics used were chloramphenicol (50 µg/mL) and tetracycline (400 ng/mL) for *S. aureus*, and ampicillin (100 µg/mL) for *E. coli* (Merck&Cie, Buchs, Switzerland).

### 2.2. Phage Susceptibility Testing (PST)

Phage susceptibility was tested using diluted drop tests (DDT). TSB soft agar with 0.75% agar was inoculated with 300 µL of bacterial overnight cultures. Serial dilutions (10^−1^ to 10^−7^) of phage suspensions were dropped on the surface of solidified soft agar. After 24 h at 37 °C, lysis zones were observed and PFU counted for phage titer determination.

### 2.3. Genomic DNA Extraction and Purification, PCR, Plasmid Construction, and Transformation

Genomic DNA extraction and purification from Laus102, 16C12, and RN4220 were performed using Nuclei Lysis Solution and Protein Precipitation Solution (Promega AG, Dübbendorf, Switzerland) following the protocol from Bae et al., 2008 [20]. The region encompassing the IS element in clone 16C12 was PCR amplified using the GoTaq DNA Polymerase (Promega AG) from purified 16C12 genomic DNA in two complementary fragments, following the manufacturer’s recommendations. The first fragment was obtained with the forward primer 5′-GAT GGT ACC GTC AAA GTG GGA GAG GTA TAA TG-3′ and the reverse primer 5′-GGT AAC TTT CTT CCA CAG TTC-3′ and the second fragment with the forward primer 5′-CGT GAA GGT GAA CAT ATG AG-3′ and the reverse primer 5′-GGC GAG CTC TAA TTT ATT TAG TGG ATA AGT GAT ATG-3′ (Microsynth AG, Balgach, Switzerland). Both fragments were sequenced from both ends using the same primers as for the PCR and assembled manually. The reconstituted fragment was compared to the core nucleotide database using blastn (https://blast.ncbi.nlm.nih.gov/Blast.cgi, accessed on 1 September 2024). Plasmid pRMC2 (Addgene, Teddington, UK) was extracted from ElectroMAX DH10B (Thermo Fisher Scientific, Waltham, MA, USA) using the QIAprep Spin Miniprep Kit (Qiagen, Hilden, Germany) according to the manufacturer’s recommendations. Extracted pRMC2 was digested with the EcoRI and SacI (New England Biolabs, Ipswich, MA, USA). Linearized pRMC2 was purified from agarose gel using the QIAquick PCR and Gel Cleanup Kit (Qiagen) according to the manufacturer’s recommendations. The *tarM* gene was amplified by PCR using the GoTaq DNA Polymerase (Promega AG) with RN4220 genomic DNA as template and primer pair 5′-CGG GAG CTC GGT AAA GGA ATA ATT ATA ATG-3′/5′-CGC GAA TTC AGG TAC TCA TGA ATA CCT AGG-3′ (Microsynth AG), following the manufacturer’s recommendations. *tarM* amplification was checked on a 1% agarose gel, and the PCR product purified using the QIAquick PCR and Gel Cleanup Kit (Qiagen), digested with EcoRI and SacI, and purified again. Ligation of pRMC2 and *tarM* to obtain pRMC2-*tarM* was performed with T4 DNA ligase (New England Biolabs) according to the manufacturer’s recommendations. To build pRMC2-*tarS*, the same method was used except that KpnI (New England Biolabs) was used instead of EcoRI for the digestion of the plasmid and the PCR product. The *tarS* gene was amplified by PCR using purified Laus102 chromosomal DNA as template with the forward primer 5′-GAT GGT ACC GTC AAA GTG GGA GAG GTA TAA TG-3′ and the reverse primer 5′-GGC GAG CTC TAA TTT ATT TAG TGG ATA AGT GAT ATG-3′ (Microsynth AG). pRMC2-*tarM* was first transformed into ElectroMAX DH10B cells. Electroporation was performed in 2 mm electroporation cuvettes (Bio-Rad, Hercules, CA, USA) at 2.5 kV using the Eppendorf Eporator (Eppendorf, Hamburg, Germany). Transformed cells were selected on LA plates containing 100 µg/mL ampicillin. A colony PCR was performed to screen colonies for the presence of pRMC2-*tarM*, using primers designed to amplify *tarM* (see above). Purified PCR products were sequenced from both ends with the same primers to verify *tarM* sequences. After verification, pRMC2-*tarM* were extracted from DH10B as described above for pRMC2 and used to transform RN4220. Electrocompetent RN4220 and Laus102 strains were freshly prepared using a protocol based on Löflblom et al., 2007 [21], except that the cells were grown in TSB. pRMC2-*tarM* was first transformed into RN4220 at 1.8 kV and transformed cells were selected on TSA plates containing 15 µg/mL chloramphenicol. Plasmids were extracted as described above, except that lysostaphin (50 µg/mL, Merck&Cie) was added to P1 buffer and samples resuspended in P1 were incubated for 30 min at 37 °C with shaking. Plasmids were checked for the correct sequence of *tarM* by PCR then sequencing of the PCR products (using the same primers as above) was performed. The plasmid pRMC2-*tarM* was further used to transform Laus102 using the same protocol and control as for RN4220. Expression of *tarM* was induced by tetracycline at 400 ng/mL. The same method was used to transform the mutants 16C02 and 16C12 with plasmid pRMC2-*tarS.*

### 2.4. In Vitro Assays

Turbidity assays: 10 µL of *S. aureus* (OD_600nm_ = 0.6 ≈ ca. 8 log_10_ CFU/mL) were added to 280 µL TSB in 96-well plates. Phage 66 was added at MOI 10. OD_600nm_ was recorded every 10 min for 24 h with 3 s shaking before reading. Experiments were performed in triplicate.

Time-kill assays: ca. 7.3 log_10_ CFU were incubated with phage 66 (MOI 10) in 20 mL TSB. At 0 h and 2 h, samples were serially diluted in PBS (pH = 3) to neutralize phage activity, then plated on TSA plates for colony counting. Experiments were performed in triplicate.

Adsorption assays: 5 mL of *S. aureus* (OD_600nm_ = 0.6 ≈ ca. 8 log_10_ CFU/mL) were mixed with phage 66 (MOI 0.01) and incubated at 37 °C for 20 min. Mixtures were centrifuged (9000× *g*, 3 min), and unadsorbed phages quantified in classical double-layer assays (DLA) with Laus102 as indicator strain. The % of adsorption was calculated from the initial and unadsorbed phage counts.

### 2.5. Whole Genome Sequencing (WGS) and Comparative Genomics

Bacterial genomic libraries were prepared with an optimized protocol and standard Illumina adapter sequences, and sequencing was performed with Illumina technology NovaSeq 6000 with read mode 2 × 150 bp paired ends (Illumina, San Diego, CA, USA) at Eurofins Genomics Germany GmbH (Ebersberg, Germany). Reads were assembled and contigs annotated using the Bacterial and Viral Bioinformatics Resource Center (BV-BRC) v.3.42.3 pipelines for bacterial genome assembly and annotation set to default parameters, respectively (https://www.bv-brc.org/, accessed on 1 January 2025). Comparative genomics was performed with the BV-BRC variation analysis tool set to default parameters and manually using blast (https://blast.ncbi.nlm.nih.gov/Blast.cgi, accessed on 1 January 2025).

### 2.6. RNA Purification, Sequencing, and Analysis

RNA purification was performed using the RNeasy Protect Bacteria Mini Kit (Qiagen). The bacterial lysis step was optimized based on [22] with DNase digestion using the RNase-Free DNase Set (Qiagen). RNA quality was assessed on a Fragment Analyzer (Agilent Technologies, Santa Clara, CA, USA). The RNAs had RNA quality number (RQN) between 9.4 and 10. RNA-Seq libraries were prepared from 500 ng of total RNA with the Illumina Truseq stranded mRNA Prep reagents (Illumina) using a unique dual indexing strategy and following the official protocol. The polyA selection step was omitted and rRNA was depleted with the QIAseq FastSelect-5 S/16 S/23 S kit (Qiagen). Libraries were quantified by Qubit (Thermo fisher Scientific) and their quality assessed on a 5200 Fragment Analyzer System (Agilent Technologies). Sequencing was performed on an Illumina NovaSeq 6000 for 100 cycles single read. Sequencing data were demultiplexed using the bcl2fastq2 Conversion Software v.2.20 (Illumina). Reads were analyzed using the BV-BRC v.3.42.3 pipeline for RNA-Seq analysis with the Tuxedo strategy. Laus102 full genomic sequence was used as target genome to align RNA reads from Laus102, 16C02, 16C12, 16C02Ω*tarS*, and 16C12Ω*tarS*, while RN4220 full genomic sequence was used as target genome to align RN4220 reads. Gene expression levels were given in FPKM (Fragments Per Kilobase Million) with a 95% confidence interval. To compare gene expression levels between the different strains, TPM (transcripts per million) was calculated from FPKM with the following formula: TPMi = (FPKMi × 10^6^/∑(FPKMj), with i representing gene i and j denoting the total number of genes.2.7. Statistical Analysis

Student’s t-tests and one-way ANOVA with Dunnett’s test were run using GraphPad Prism v10.4.1 (www.graphpad.com).

## 3. Results

### 3.1. Resistance to the Podovirus, but Not to the Myovirus or Flucloxacillin, Was Selected in Rat Vegetations Treated with a Flucloxacillin/Two-Phage Cocktail Combination

As previously reported [17], treatment of experimental MSSA infective endocarditis in rats with flucloxacillin combined with a two-phage cocktail composed of the podovirus *Routneeviridae* phage 66 and the myovirus *Herelleviridae* phage vB_SauH_2002 (phage 2002) led to in vivo selection of resistance to phage 66. While most of the recovered clones isolated from vegetations of the most infected animal (60 in total) were still susceptible to both phages (Appendix A), DDT confirmed that two clones, 16C02 and 16C12, were fully resistant to phage 66 while retaining full susceptibility to phage 2002 (Appendix A).

### 3.2. Mutations in tarS as the Likely Genetic Basis for In Vivo Podovirus Resistance

Whole-genome sequencing and comparative genomics of clones 16C02, 16C12, a susceptible clone 18C10, and WT strain Laus102 revealed distinct mutations in *tarS* in both resistant isolates. In 16C02, an IS6-like IS257 transposase insertion disrupted *tarS*, splitting the gene across two loci (ACJ0H2_07840 and ACJ0H2_13115). This IS257 insertion was confirmed by the sequencing of two complementary PCR fragments, which reconstituted the missing region containing the IS element (Appendix A). Clone 16C12 harbored an 11-nucleotide deletion resulting in a frameshift and premature stop codon (M295fs) (Figure 1). Other mutations were shared between 16C02, 16C12, and the susceptible clone 18C10 and were thus unlikely to contribute to the resistance phenotype (Appendix A). The crystal structure of TarS is known and reveals that the 573 amino acids long protein is composed of three main domains: a catalytic domain (residues 1–349), which includes a C-terminal helical bundle (residues 217–319); a linker region (residues 320–352); and a trimerization domain (residues 353–573), which is essential for enzymatic activity [23]. The 11-nucleotide deletion results in a frameshift and a premature stop codon at methionine 295 (M295fs), producing a truncated protein lacking part of the catalytic domain, the entire linker region, and the trimerization domain. This truncation likely renders the protein non-functional.

The IS257 insertion occurs at amino acid position 257, which lies within the catalytic domain. Although the insertion does not disrupt residues directly involved in catalysis or hydrogen bonding, it may interfere with proper folding or trimerization of TarS, potentially impairing its enzymatic function.

### 3.3. Expression Levels of tarS Did Not Account for Resistance

Despite the mutations, *tarS* expression was still detectable in the resistant clones. Transcript levels measured by RNA-seq were 172.12 ± 6.91 TPM in Laus102, 96.59 ± 4.06 TPM in 16C02, and 199.29 ± 6.91 TPM in 16C12 (Figure 2), suggesting that resistance was not due to transcriptional silencing but rather to functional inactivation of TarS.

### 3.4. Complementation with Wild-Type tarS Restored Phage 66 Susceptibility

To confirm the role of *tarS* mutations in the resistance phenotype, the Laus102 WT *tarS* gene was cloned under a tetO promoter in plasmid pRMC2 and introduced into clones 16C02 and 16C12, generating 16C02Ω*tarS* and 16C12Ω*tarS*, respectively. Complemented strains expressed *tarS* at high levels (751.44 ± 12.40 TPM in 16C02Ω*tarS* and 537.67 ± 10.85 TPM in 16C12Ω*tarS*) (Figure 2). Phage 66 formed plaques on the WT strain and both complemented mutants but not on the parental resistant strains (Figure 3A). Turbidity assays showed a 6–8 h growth delay in the presence of phage 66 for Laus102, 16C02Ω*tarS*, and 16C12Ω*tarS*, but not for the uncomplemented resistant mutants (Figure 3B,C). Of note, growth rates were comparable in all strains in the absence of phages. In time-kill assays, phage 66 significantly reduced log_10_ CFU/mL in Laus102 from 6.14 ± 0.13 to 0.70 ± 0.68 at 2 h (*p* < 0.01). No reduction was observed in 16C02 and 16C12, where bacterial counts increased slightly to 6.86 ± 0.12 and 6.86 ± 0.14 log10 CFU/mL, respectively. Complementation restored partial susceptibility in 16C02Ω*tarS* (6.58 ± 0.34 to 4.83 ± 0.13, *p* < 0.01) and near-complete susceptibility in 16C12Ω*tarS* (6.36 ± 0.35 to 1.60 ± 0.21, *p* < 0.001) (Figure 4).

### 3.5. Resistance Correlated with Impaired Phage Adsorption

Adsorption assays demonstrated that phage 66 efficiently adsorbs to Laus102 (96.93 ± 0.83%) but poorly to 16C02 (56.00 ± 4.00%, *p* < 0.01) and 16C12 (54.67 ± 11.55%, *p* < 0.05). Complemented strains exhibited restored adsorption levels (97.73 ± 0.61% for 16C02Ω*tarS* and 97.47 ± 0.61% for 16C12Ω*tarS*) (Figure 5).

### 3.6. Analysis of tarM and tarP Sequences

Genomic analysis showed that *tarM* was identical in Laus102, 16C02, 16C12, and 18C10, with a single SNP (E266G) differentiating RN4220 (Appendix A). Similarly to RN4220 [24], *tarP* was not identified on the genomes of Laus102 and derivatives.

### 3.7. Low tarM Expression Likely Explains Podovirus Innate Susceptibility of Laus102

Phage 66 did not form PFUs on RN4220 (Appendix A), consistent with previous findings that TarM-mediated α-GlcNAcylation blocks podovirus adsorption [14]. Although Laus102 harbored the same *tarM* sequence (except for SNP E266G), RNA-seq revealed > 10-fold lower *tarM* expression in Laus102 compared to RN4220 (9.00 ± 1.81 TPM vs. 100.64 ± 5.15 TPM, respectively, *p* < 0.0001). Of note, *tarS* expression was only slightly lower in Laus102 than RN4220 (172.12 ± 6.91 TPM vs. 232.94 ± 7.49 TPM, respectively, *p* < 0.01) (Figure 6).

## 4. Discussion

In a prior study, rats with MSSA infective endocarditis were treated with a phage cocktail (vB_SauH_2002 + phage 66) and low-dose flucloxacillin. Despite strong phage/antibiotic synergy, some rats remained infected after 24 h [17], with some recovered clones being resistant to podovirus phage 66. Here, we sequenced and characterized two such resistant clones. Comparative genomics revealed that *tarS* was disrupted—by a transposon in 16C02 and an indel in 16C12—likely resulting in non-functional TarS. RNA-seq confirmed *tarS* expression was still detectable, suggesting impaired function rather than transcriptional silencing. TarM and TarS catalyze α- and β-GlcNAc WTA glycosylation at the C4 position of Rbo-P, while TarP adds β-GlcNAc at C3 [23,25,26]. The *tarP* gene, absent in Laus102, is typically carried by temperate phages in HA-MRSA or LA-MRSA [23]. TarS-mediated β-GlcNAcylation enables podovirus infection, while TarM-derived α-GlcNAc blocks it [14,26]. Complementation of the *tarS* mutants with WT *tarS* from Laus102 (via pRMC2) restored phage 66 susceptibility, as confirmed by strong *tarS* expression and regained adsorption and lytic activity in DDT, turbidity, and time-kill assays. These findings confirm TarS is essential for phage 66 infection of Laus102 in vivo, especially in blood. Surprisingly, Laus102 had innate susceptibility to phage 66 despite harboring *tarM*. RNA-seq revealed *tarM* expression was >10 times lower in Laus102 than RN4220, possibly explaining this counterintuitive susceptibility. Although the promoter regions of *tarM* in Laus102 and RN4220 are identical, its expression was significantly lower in Laus102. The only sequence difference identified is an SNP resulting in an E266G substitution in the TarM protein. Given the identical upstream regulatory regions, this SNP may influence mRNA stability directly, although this remains to be experimentally verified. In addition, the inspection of the TarM crystal structure suggests that E266 is involved in forming a stabilizing hydrogen bond. Substitution with glycine may disrupt local structure and impair trimer formation or enzymatic activity, which are both essential for proper TarM function [27]. Therefore, the E266G substitution could have a dual impact—reducing RNA stability and/or translation efficiency, as well as directly impairing the enzyme’s structural integrity and function. Although attempts to overexpress RN4220 *tarM* in Laus102 failed, these combined effects may explain the high susceptibility of Laus102 to phage 66 despite carrying a *tarM* gene. Li et al. demonstrated that in a *tarM*^+^/*tarS*^+^ genetic background, TarM activity predominates over TarS, resulting in α-GlcNAcylation of WTA and conferring resistance to podoviruses [14]. Our results support this observation insofar as *tarS* was expressed twice as much as *tarM* in RN4220.

Our study showing the natural selection of *tarS*-deficient mutants under podovirus pressure in vivo provide a compelling biological proof-of-concept that TarS can be targeted not only pharmacologically but also through phage therapy. Interestingly, previous reports have identified TarS as a promising target whose inhibition can re-sensitize MRSA to β-lactam antibiotics [23]. Accordingly, podovirus/β-lactam combination may represent a dual-action strategy against MRSA: podoviruses actively selecting for mutants with defective WTA glycosylation, which, although phage-resistant, become resensitized to β-lactam antibiotics. This phenomenon would parallel known evolutionary trade-offs, wherein the acquisition of resistance to one selective pressure inadvertently confers sensitivity to another, such as antibiotic resistance and fitness cost [28]. In this context, podoviruses would act as both selective agents and potential adjuvants in β-lactam-based therapies. Given that several podoviruses capable of infecting MRSA strains have already been isolated, this novel strategy to manage MRSA infections warrants further and detailed investigation.

## Figures and Tables

**Figure 1 viruses-17-01039-f001:**
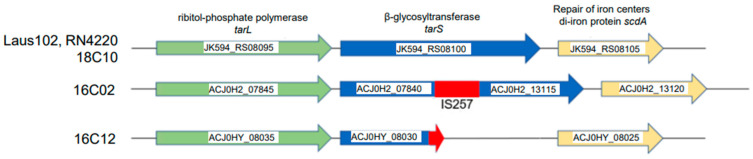
Schematic representation of *tarS* and its flanking genes carried by the genomes of the studied clones. Locus tags for strains Laus102, 16C02, and 16C12 and functions of the encoded proteins are indicated.

**Figure 2 viruses-17-01039-f002:**
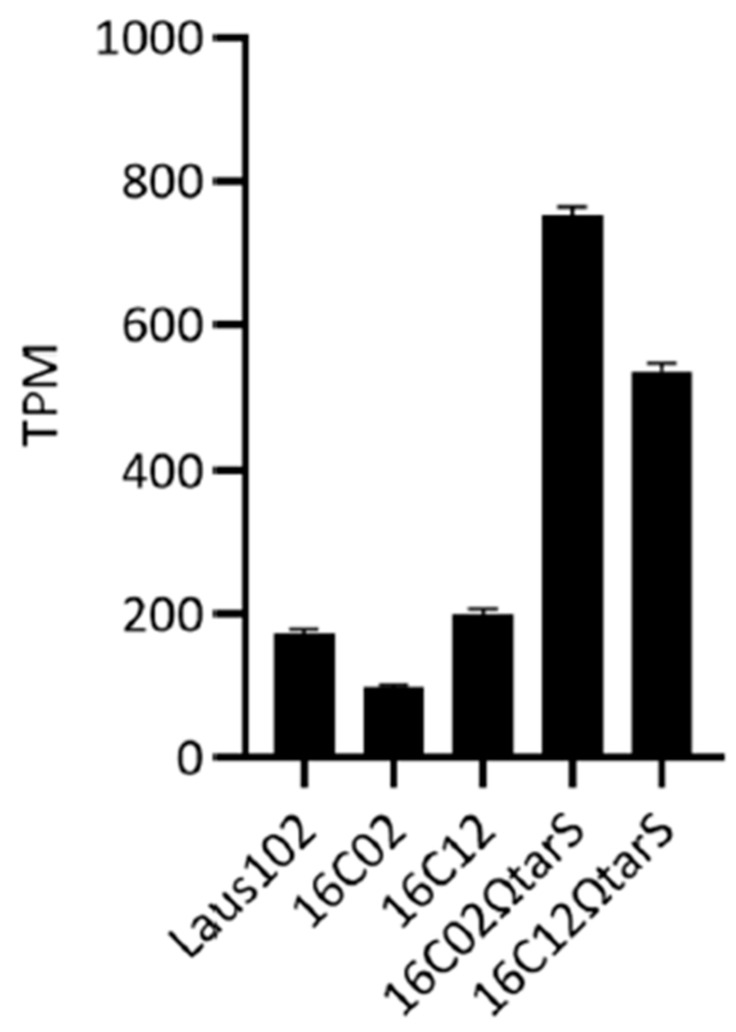
Levels of *tarS* expression in selected *S. aureus* strains. TPM: transcript per million.

**Figure 3 viruses-17-01039-f003:**
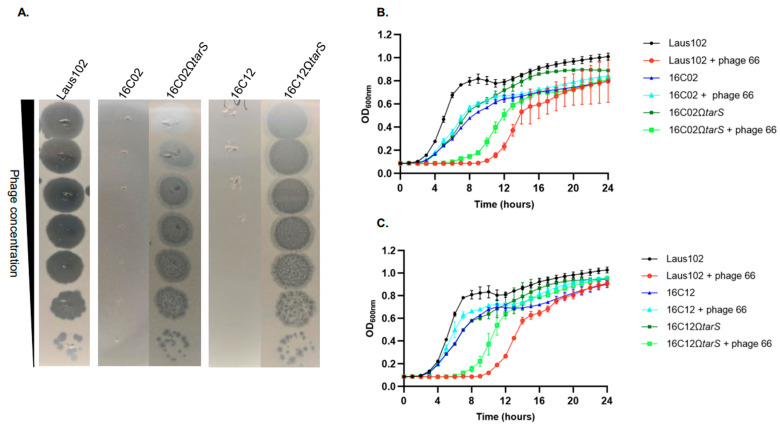
(**A**) Diluted drop tests of phage 66 on Laus102, 16C02, 16C12 and their respective complemented mutants 16C02Ω*tarS* and 16C02Ω*tarS*. Turbidity assays of phage 66 at MOI = 10 on (**B**) Laus102, 16C02, and its respective complemented mutant 16C02Ω*tarS* and on (**C**) Laus102, 16C12 and its complemented mutants 16C12Ω*tarS*. MOI, multiplicity of infection. Each dot represents the mean ± SD of triplicates.

**Figure 4 viruses-17-01039-f004:**
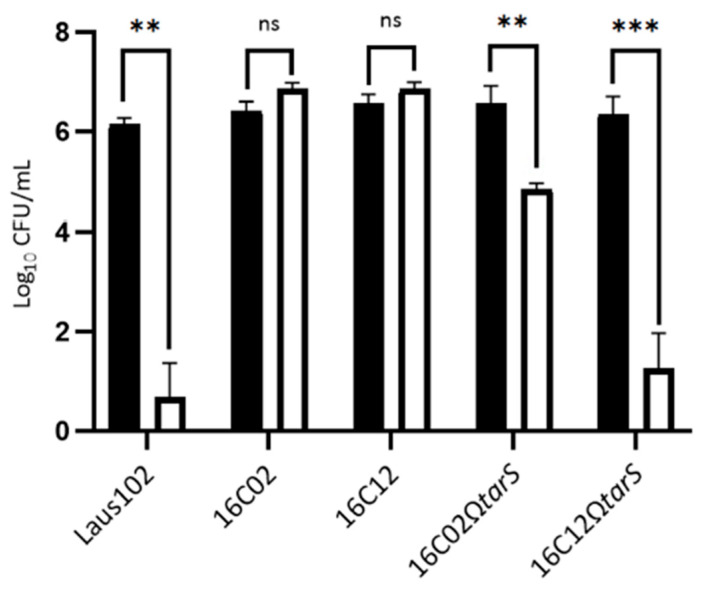
Time-kill assays with *S. aureus* isolates and phage 66 at an MOI of 10. Filled and open histograms indicate bacterial loads (Log10 CFU/mL) 0 h and 2 h post phage 66 challenge, respectively. MOI, multiplicity of infection. Each experiment has been performed in triplicates. ***. *p* < 0.001; **. *p* < 0.01; ns. not significant (Student’s *t*-test).

**Figure 5 viruses-17-01039-f005:**
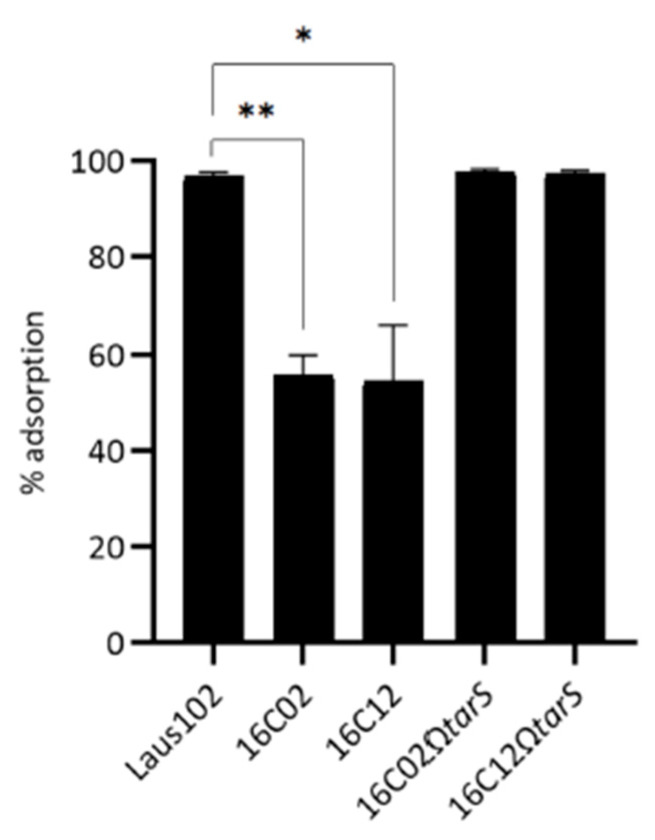
Adsorption tests of phage 66 on indicated strains. *. *p* < 0.05; **. *p* < 0.01 (Student’s *t*-test).

**Figure 6 viruses-17-01039-f006:**
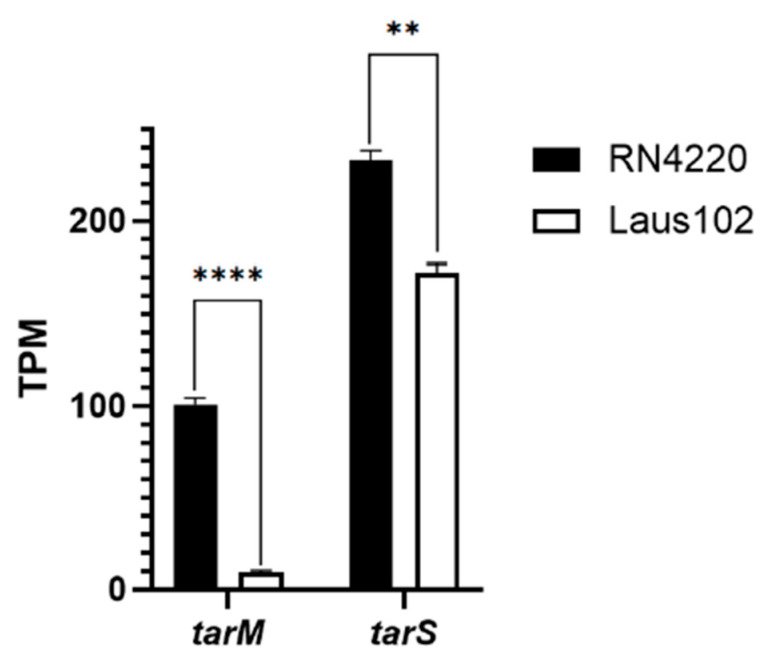
mRNA expression of *tarM* and *tarS* in RN4220 and Laus102. TPM: transcripts per million. Each experiment has been performed in triplicates. ****. *p* < 0.0001; **. *p* < 0.01 (Student’s *t*-test).

## Data Availability

WGS and sequencing reads for Laus102, 18C10, 16C02, and 16C12 are in NCBI BioProject PRJNA689979 under accession numbers (WGS/SRA): JAETXI000000000/SRR33651905, JAETWS000000000/SRR33652073, JBJLKP000000000/SRR31513294, JBJLKQ000000000/SRR31513293, respectively. The WGS project of RN4220 has accession number AFGU01000000. RNA-Seq reads are accessible with the SRA accession number SRX28881584 (Laus102), SRX28881585 (16C02), SRX28881586 (16C12), and SRX28882349 (RN4220).

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
