# Peer review of "In Vivo Emergence of Podovirus Resistance via tarS Mutation During Phage-Antibiotic Treatment of Experimental MSSA Endocarditis"

_viruses, 2025, doi:10.3390/v17081039_

Round 1

Reviewer 1 Report

Comments and Suggestions for Authors

This manuscript describes the mechanistic basis for phage resistance which has developed in vivo following combined treatment, in a rat model of cardiac MSSA infection, with a two phage cocktail and an antibiotic. Bacteria were isolated from the host, and a subset of these isolates shown to be resistant to one of the two phage.

The work combines genomic sequencing, complementation, assays for phage infectivity and RNA analysis to show that tarS and tarM genes are important for determining phage susceptibility/resistance in this in vivo setting.

The work is carefully designed and performed.

 I had only a few minor suggestions, mostly around interpretation of the effects of the observed mutations:

Section 3.2. Sequencing of two phage 66 resistant strains revealed changes (an IS257 insertion) and an 11 nucleotide deletion resulting in frameshift and premature stop. Given that the crystal structure of TarS is known, I suggest that the authors explicitly state the impact of the mutations on the TarS protein eg. for the truncation at M295 – how many amino acids long is the wild type protein, and therefore what fraction of the wild type protein is produced; in which domain of the protein (an enzyme) does the truncation occur?

Section 3.7 TarM

A second question which arose is why the parental strain (Laus102) is susceptible to podophage 66 at all, since tarM activity would normally prevent podophage infection.

There were no differences seen in the promoter regions of Laus102 and RN4220, which is phage 66 resistant. Only a single base change within TarM was observed, which implies it is involved in the large difference in transcript levels of this gene between the two strains (Fig. 6). The general statement (first paragraph of discussion) that “an E226G SNP within TarM might impact its expression or function” should be briefly expanded. If expression levels are very much lower, but the promoter sequences are identical, doesn’t this result imply that the SNP impacts RNA stability directly? The fact that E266G may impact enzyme activity seems a separate question, unless TarM for example has some unknown mechanism of self-regulation at the transcriptional level. The authors could also look more closely here at whether the E266G mutation is likely to effect enzyme activity – TarM crystal structure is known; a quick look at the PDB suggests that the change would result in loss of a hydrogen bond, likely destabilising the enzyme, which works as a trimer.

The observed phenotype could be a combined effect as a result of both lower mRNA levels and reduced activity.

Attempts to overexpress RN4220 TarM in Laus102 failed (first paragraph discussion). Did the authors try to overexpress the E266G mutant or use some other positive control here?

Figure 6 legend. Reference to MOI not required.

Supp table 1.

It is difficult to distinguish the italic (shared variations) from the non-italic fonts in the case of the single base changes.

Author Response

Comment 1: This manuscript describes the mechanistic basis for phage resistance which has developed in vivo following combined treatment, in a rat model of cardiac MSSA infection, with a two phage cocktail and an antibiotic. Bacteria were isolated from the host, and a subset of these isolates shown to be resistant to one of the two phage.

The work combines genomic sequencing, complementation, assays for phage infectivity and RNA analysis to show that tarS and tarM genes are important for determining phage susceptibility/resistance in this in vivo setting.

The work is carefully designed and performed.

 I had only a few minor suggestions, mostly around interpretation of the effects of the observed mutations:

Section 3.2. Sequencing of two phage 66 resistant strains revealed changes (an IS257 insertion) and an 11 nucleotide deletion resulting in frameshift and premature stop. Given that the crystal structure of TarS is known, I suggest that the authors explicitly state the impact of the mutations on the TarS protein eg. for the truncation at M295 – how many amino acids long is the wild type protein, and therefore what fraction of the wild type protein is produced; in which domain of the protein (an enzyme) does the truncation occur?

Response 1: Thank you for this insightful comment. We have added the following text to clarify the implications of the identified mutations at lines 210-221:

“The crystal structure of TarS is known and reveals that the 573 amino acids long protein is composed of three main domains: a catalytic domain (residues 1–349), which includes a C-terminal helical bundle (residues 217–319); a linker region (residues 320–352); and a trimerization domain (residues 353–573), which is essential for enzymatic activity (23). The 11-nucleotide deletion results in a frameshift and a premature stop codon at methionine 295 (M295fs), producing a truncated protein lacking part of the catalytic domain, the entire linker region, and the trimerization domain. This truncation likely renders the protein non-functional.
The IS257 insertion occurs at amino acid position 257, which lies within the catalytic domain. Although the insertion does not disrupt residues directly involved in catalysis or hydrogen bonding, it may interfere with proper folding or trimerization of TarS, potentially impairing its enzymatic function.”

Comment 2: Section 3.7 TarM

A second question which arose is why the parental strain (Laus102) is susceptible to podophage 66 at all, since tarM activity would normally prevent podophage infection.

There were no differences seen in the promoter regions of Laus102 and RN4220, which is phage 66 resistant. Only a single base change within TarM was observed, which implies it is involved in the large difference in transcript levels of this gene between the two strains (Fig. 6). The general statement (first paragraph of discussion) that “an E226G SNP within TarM might impact its expression or function” should be briefly expanded. If expression levels are very much lower, but the promoter sequences are identical, doesn’t this result imply that the SNP impacts RNA stability directly? The fact that E266G may impact enzyme activity seems a separate question, unless TarM for example has some unknown mechanism of self-regulation at the transcriptional level. The authors could also look more closely here at whether the E266G mutation is likely to effect enzyme activity – TarM crystal structure is known; a quick look at the PDB suggests that the change would result in loss of a hydrogen bond, likely destabilising the enzyme, which works as a trimer.

The observed phenotype could be a combined effect as a result of both lower mRNA levels and reduced activity.

Response 2: We thank the reviewer for this excellent observation and the thoughtful interpretation. We have now revised the discussion to elaborate on the possible effects of the E266G SNP within TarM. Another explanation we added is the possible different activity of the promoter of tarM in between the two strains. Here is the text that was added at line 312-323:

“Although the promoter regions of tarM in Laus102 and RN4220 are identical, its expression was significantly lower in Laus102. The only sequence difference identified is a SNP resulting in an E266G substitution in the TarM protein. Given the identical upstream regulatory regions, this SNP may influence mRNA stability directly, although this remains to be experimentally verified. In addition, the inspection of the TarM crystal structure suggests that E266 is involved in forming a stabilizing hydrogen bond. Substitution with glycine may disrupt local structure and impair trimer formation or enzymatic activity, which are both essential for proper TarM function (27). Therefore, the E266G substitution could have a dual impact—reducing RNA stability and/or translation efficiency, as well as directly impairing the enzyme’s structural integrity and function. Although attempts to overexpress RN4220 tarM in Laus102 failed, these combined effects may explain the high susceptibility of Laus102 to phage 66 despite carrying a tarM gene”

Comment 3: Attempts to overexpress RN4220 TarM in Laus102 failed (first paragraph discussion). Did the authors try to overexpress the E266G mutant or use some other positive control here?

Response 3: We thank the reviewer for this pertinent question. In this study, we focused our overexpression attempts in Laus102 of the wild-type tarM allele from RN4220, which is responsible for RN4220 resistance to phage 66. This attempt was done to investigate the importance of the E266G mutation in the TarM activity. We did not try to overexpress Laus102 allele or another positive control.

Comment 4: Figure 6 legend. Reference to MOI not required.

Response 4: It was removed.

Comment 5: Supp table 1.

It is difficult to distinguish the italic (shared variations) from the non-italic fonts in the case of the single base changes.

Response 5: We changed the way to present the different variations in Figure S1 as now indicated in the table title: “Variations that are not shared between strains are indicated in bold. The 11 nucleotides indel leading to the Met295fs in tarS is indicated in bold red for clone 16C12. All other variations are shared between either 16C02, 16C12, or 18C10 and Laus102”.

Reviewer 2 Report

Comments and Suggestions for Authors

The authors analyzed resistant S. aureus mutants previously isolated from cardiac vegetations in rats with MSSA infective endocarditis. The results showed the tarS-deficient mutants are naturally selected under podovirus pressure in vivo and  TarS can be targeted through phage therapy. As TarS is known to be a promising target for re-sensitizing MRSA to β-lactam antibiotics, the results of this study may lead to further research on the effectiveness of combining phage and β-lactam antibiotics in the treatment of MRSA.

Author Response

We thank Reviewer 2 for the thorough review and for the appreciation of our work.

Reviewer 3 Report

Comments and Suggestions for Authors

Cherbuin et al. present a refreshingly well written manuscript elucidating the adsorption mechanism of phage 66, a podovirus targeting Staphylococcus aureus. The authors convincingly demonstrate that the β-GlcNAcylation of wall teichoic acid molecules, executed by TarS, allows phage 66 to recognize the Staphylococcus aureus cell, while α- GlcNAcylation of the same site in the wall teichoic acid, executed by TarM, blocks phage 66 adsorption. In addition, the authors make a convincing case that such findings are crucially important to understand how bacteria can adapt to selective pressure exerted by bacteriophages and that a better understanding of such phenomena will allow us to design better phage therapy applications.

Specific comments:

The lack of line numbers makes the task of the reviewer difficult.

Page 1, Abstract, lines 4-6: “Here we show that resistant clones harbor mutations in tarS, which encodes a glycosyltransferase essential for β-GlcNAcylation of wall teichoic acid (WTA), a key step for phage adsorption.” Too much or too little information, condensed in one sentence disturbs the language logic. What exactly is the key step, that a resistant clone harbors, that tarS encodes or the enzymatic action of TarS? And, should it not be “ … for phage 66 adsorption” for the last part of the sentence to be correct? Already, vB_SauH_2002 does not seem to care very much about the presence of the β-GlcNAC moiety in the WTA. Better end the first sentence after “ … of wall teichoic acid (WTA).” and reformulate the last part separately.

Page 1, Abstract, line 8: “ … restored by tarS complementation …”. As in the discussion, please insert a WT here. “ … restored by WT tarS complementation“.

Page 3, Material and Methods, line 24: “tarS gene was …” better “The tarS gene was …” in order not to capitalize the gene.

Page 3, Material and Methods, line 29: “ … at 2.5kV using …”. Please insert a space between the number and the unit. Please also in line 37, “at 1.8kV“, line 39, “50μg/mL“ and page 4, line 18, “150bp”.

Page 5, Figure1: Figure 1 shows a schematic representation of the genes tarL, tarS and scdA. This should not only be reflected in the figure legend, but also within the figure. “ribitol-phospate-polymerase tarL” (not the protein TarL), “ß-ribosyltransferase tarS” and “repair of iron centers di-iron protein scdA”. Alternatively, the authors will have to mention in the figure that the genes encode the respective proteins. The same argument is also valid for figure S4.

Page 9, Discussion, line 9: “tarP, absent in Laus102 …” better “The tarP gene, …” in order not to capitalize the gene.

Figures S1 and S3. The authors define their “diluted drop test” as (DDT) in the material and method section of the manuscript. To redefine DDT in both supplemental figures is redundant.

Author Response

We thank Reviewer 3 for his/her thorough review and the insightful comments provided. These have been carefully considered and addressed as follows in the revised version of our manuscript:

Comment 1: The lack of line numbers makes the task of the reviewer difficult.

Response 1: We fully agree with this comment, as we encountered the same issue during our own revisions. We did submit an initial version that included line numbering. Therefore, we do not know why this numbering was missing from the version made available to the reviewer. The submitted revised version also includes line numbering one our end.

Comment 2: Page 1, Abstract, lines 4-6: “Here we show that resistant clones harbor mutations in tarS, which encodes a glycosyltransferase essential for β-GlcNAcylation of wall teichoic acid (WTA), a key step for phage adsorption.” Too much or too little information, condensed in one sentence disturbs the language logic. What exactly is the key step, that a resistant clone harbors, that tarS encodes or the enzymatic action of TarS? And, should it not be “ … for phage 66 adsorption” for the last part of the sentence to be correct? Already, vB_SauH_2002 does not seem to care very much about the presence of the β-GlcNAC moiety in the WTA. Better end the first sentence after “ … of wall teichoic acid (WTA).” and reformulate the last part separately.

Response 2: The sentence has been splitted in two as suggested at Lines 20-21: "Here we show that resistant clones harbor mutations in tarS, which encodes a glycosyltransferase essential for β-GlcNAcylation of wall teichoic acid (WTA). This WTA modification has been described in vitro as critical for podoviruses adsorption.". We also precised "... in resistant clones" at line 22 and "...also in vivo..." at line 28. 

Comment 3: Page 1, Abstract, line 8: “ … restored by tarS complementation …”. As in the discussion, please insert a WT here. “ … restored by WT tarS complementation“.

Response 3: WT has been added as suggested at line 23.

Comment 4: Page 3, Material and Methods, line 24: “tarS gene was …” better “The tarS gene was …” in order not to capitalize the gene.

Response 4: This has been corrected has suggested at line 114. As similar issue has been corrected at line 105 "The tarM gene...".

Comment 5: Page 3, Material and Methods, line 29: “ … at 2.5kV using …”. Please insert a space between the number and the unit. Please also in line 37, “at 1.8kV“, line 39, “50μg/mL“ and page 4, line 18, “150bp”.

Response 5: This has been corrected as suggested at lines 119, 127, 129, and 153.

Comment 6:Page 5, Figure1: Figure 1 shows a schematic representation of the genes tarL, tarS and scdA. This should not only be reflected in the figure legend, but also within the figure. “ribitol-phospate-polymerase tarL” (not the protein TarL), “ß-ribosyltransferase tarS” and “repair of iron centers di-iron protein scdA”. Alternatively, the authors will have to mention in the figure that the genes encode the respective proteins. The same argument is also valid for figure S4.

Response 6: The figure 1 and S4 annotations have been adjusted accordingly. 

Comment 7: Page 9, Discussion, line 9: “tarP, absent in Laus102 …” better “The tarP gene, …” in order not to capitalize the gene.

Response 7: This has been changed accordingly at line 300.

Comment 8: Figures S1 and S3. The authors define their “diluted drop test” as (DDT) in the material and method section of the manuscript. To redefine DDT in both supplemental figures is redundant.

Response 8: We thank the reviewer for his/her vigilance. The definition of DDT has been removed in both Figures as as adviced.